# LLaVA-Ultra: Large Chinese Language and Vision Assistant for Ultrasound

## ABSTRACT

Multimodal Large Language Model (MLLM) has recently garnered significant attention as a prominent research focus. By harnessing the capability of powerful Large Language Model (LLM), it facilitates the transition of conversational generative AI from unimodal text to performing multimodal tasks. This blooming development has begun to significantly impact the medical field. However, visual language models in the general domain lack sophisticated comprehension required for medical visual conversations. Even some models specifically tailored for the medical domain often produce answers that tend to be vague and weakly related to the visual contents. In this paper, we propose a fine-grained and adaptive visual language model architecture for Chinese medical visual conversations through parameter-efficient tuning. Specifically, we devise a fusion module with fine-grained vision encoders to achieve enhancement for subtle medical visual semantics. Then we note data redundancy that is common in medical scenes but ignored in most prior works. In cases of a single text paired with multiple figures, we utilize weighted scoring with knowledge distillation to adaptively screen valid images mirroring text descriptions. For execution, we leverage a large-scale Chinese ultrasound multimodal dataset obtained first-hand from the hospital database. We create instruction-following data based on text derived from doctors, which ensures professionality and thus contributes to effective tuning. With enhanced architecture and quality data, our **L**arge Chinese **L**anguage **a**nd **V**ision **A**ssistant for **Ultra**sound (LLaVA-Ultra) shows strong capability and robustness to medical scenarios. On three medical visual question answering datasets, LLaVA-Ultra surpasses previous state-of-the-art models on various metrics.

## CCS CONCEPTS

• **Computing methodologies** → *Visual content-based indexing and retrieval*; *Matching*; *Image representations*.

## KEYWORDS

Multimodal large language model, Medical instruction tuning

## 1 INTRODUCTION

Generative pretraining has demonstrated effective for visual language modeling in the general domain, utilizing image-text multimodal data, as exemplified by GPT-4 [24] and LLaVA [21]. Through

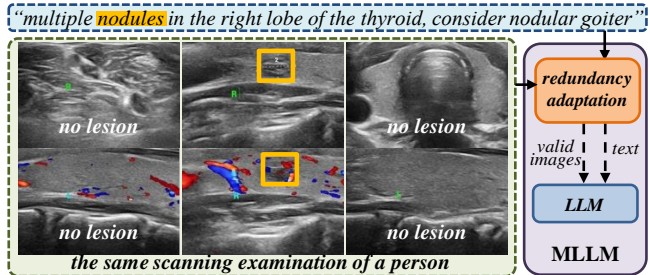

**Figure 1: Data redundancy common in medical scenes puts a need for fine-grained perception and adaption in MLLM.**

self-supervised finetuning with instruction-following data, pretrained Large Language Models (LLMs) [3, 34, 52] can be adapted to unseen tasks, resulting in improved zero-shot performance. This simple yet powerful method has extended to the multimodal field. It gives rise to Multimodal Large Language Models (MLLMs) [1, 11, 45] that excel in visual language tasks such as image reasoning and further forms user-oriented multimodal conversation assistants.

Based on the success of MLLMs in the general domain, similar initiatives have emerged for medical applications [2, 28, 50]. Current research is gradually transitioning from unimodal text to incorporating multimodal medical data. However, despite their effectiveness in general tasks, MLLMs often struggle in medical contexts. This results in inaccuracy or refusal to provide answers when faced with medical questions. It is partially due to the scarcity and difficulty in accessing parallel medical image-text data, unlike the diverse internet data available for the general domain. LLaVA-Med [17] alleviates this issue to some extent by creating multimodal medical instruction-following data for finetuning. However, it still struggles to provide more detailed correct responses. Sometimes its answers tend to be vague, focusing more on medical concepts in textual questions rather than in-depth image analysis. From the data perspective, it utilizes image-text data extracted from PubMed public papers, which may be coarser and less cross-modal matched than those from primary sources. Notably, it ignores data redundancy common in clinics shown in Fig. 1 as many other previous works. This puts a need for model enhancement and adaption for the fine-grained medical domain. Additionally, there is little exploration of extensive Chinese data in the medical multimodal field.

In this paper, we propose a **L**arge Chinese **L**anguage **a**nd **V**ision **A**ssistant for **Ultra**sound (LLaVA-Ultra), an end-to-end trained medical multimodal chatbot. To our knowledge, it is the first attempt to extend multimodal finetuning to the Chinese medical domain based on a large-scale dataset. Its domain-specific pretraining has shown effectiveness for medical vision-language (VL) tasks. Specifically, our paper makes the following contributions:

• **Enhancement for medical adaptation**. To meet the needs of subtle medical images, we leverage an extra fine-grained Segment

Anything Model (SAM) [16] encoder to jointly extract visual semantics with the CLIP encoder. A followed fusion module can effectively integrate these two typical features and thus achieve visual enhancement for better multimodal alignment. Moreover, for data redundancy common in medical scenarios, we design an adaptive sampling module with weighted scoring and knowledge distillation to automatically screen valid information. It improves the model's robustness and thus ensures correct responses in complicated practical medical scenarios.

- **High-quality medical parallel data.** We present a novel data sourcing pipeline to collect a large-scale Chinese ultrasound multimodal dataset first-hand from the hospital database. It covers professional content provided by doctors for ultrasound examinations of many body parts. We sample pairs of around 170k ultrasound images and 20k clinical texts and generate multi-modal instructions with GPT-3.5 [23] for medical instruction-tuning.
- **LLaVA-Ultra**. Contributed by the robust architecture as well as fine-grained professional data, LLaVA-Ultra shows the best practice in the Chinese medical domain. Trained in only 60 hours with 4 48GB A40s, it provides detailed answers relevant to visual content in medical conversations. On our ultrasound hospital dataset and public medical visual question answering (VQA) datasets, LLaVA-Ultra outperforms prior state-of-the-art (SOTA).

## 2 RELATED WORK

Research of Multimodal Large Language Models (MLLMs) [4, 41, 45] is an emerging hotspot currently. The key concept is to leverage pre-trained Large Language Models (LLMs) to incorporate information from other modalities, such as vision, for performing multimodal tasks like video understanding [31, 51] and embodied agent [37, 53–55]. Remarkable works such as BLIP-2 [18], LLaVA [21] and LLAMA-Adapter [48] demonstrate prominent generative capabilities including visual question answering and image captioning [8, 9]. These advancements have expanded the research of AI into new fields and hold promising prospects for further development [22].

**Medical multimodal chatbots.** Inspired by the aforementioned successes in the general domain, explorations into MLLMs have progressively transitioned into the medical domain, exemplified by models like LLaVA-Med [17] and Med-PaLM M [35]. They utilize medical datasets to perform instruction tuning for MLLMs initialized from the general domain and thus create potential applications in medical scenarios, such as medical VQA. For instance, LLAVA-Med utilizes instruction-tuning data [27] generated from the PMC-15M [49] dataset to train a multimodal medical chatbot. Consequently, it extends LLaVA to the medical domain.

Although LLaVA-Med and our proposed LLaVA-Ultra share a similar base model LLaVA, they differ significantly: (*i*) *Model architecture.* LLaVA-Med is based on the base model LLaVA without significant modifications. However, we propose effective enhancements to the model structure to adapt the characteristics of medical data. It primarily focuses on improving comprehension of subtle visual semantics and adapting to data redundancy in medical scenarios. Thus, our LLaVA-Ultra improves effectiveness and robustness in medical applications. (*ii*) *Data source and characteristic.* LLaVA-Med relies on rougher internet-based data PMC-15M [49], whereas our model utilizes professional detailed data first-hand sourced from the hospital. It incorporates data redundancy and

data similarity and is more relevant to real-world healthcare scenarios, placing higher requirements on modeling capabilities than the former. LLaVA-Med's data is in English, while ours is in Chinese. Other commonly used datasets such as MedICaT [32], ROCO [26] and MIMIC-CXR [14] exhibit similar limitations as above.

**Chinese medical chatbots.** Medical assistants are emerging in the Chinese language field as well. Notably, BenTsao [38] integrates Chinese medical knowledge bases into both the training and inference phases of LLMs, culminating in the development of a medical chatbot in the Chinese language. This approach mirrors the construction method adopted by most existing Chinese medical chatbots, such as MedicalGPT [44], HuatuoGPT [47], DoctorGLM [43] and XrayGLM [40]. However, to the best of our knowledge, XrayGLM stands as the only existing multimodal Chinese medical chatbot capable of processing image inputs. Nevertheless, it relies on relatively coarse data and primarily focuses on chest X-rays, thus lacking the breadth and diversity of data. Meanwhile, the presented model response examples are diagnostics mainly for images without lesions, lacking results for diverse medical images. And it fails to produce more detailed answers. In contrast, our proposed model, LLaVA-Ultra, offers substantial enhancements in these respects.

## 3 MULTIMODAL CONVERSATIONAL MODEL IN CHINESE MEDICAL DOMAIN

### 3.1 Ultrasound Concept Feature Alignment

We employ a network architecture that is based on the multimodal conversation model LLaVA, incorporating a projection module to link the visual encoder with the language model. The model parameters are initialized with weights from LLaVA-Med in the English medical domain, followed by finetuning [39] using medical domain instructions derived from our Chinese ultrasound dataset. For each paired sample, given textual instructions $X_q$ and image inputs $X_v$, we ask the model to produce answers $X_a$ related to the original caption $X_c$, such as giving a diagnosis or describing visual contents. The instruction tuning process can be formulated as follows,

$$p(X_a|X_c, X_v, X_q) = \prod_{l=1}^{L} p_\theta(x_l|X_c, X_v, X_q, x_{1:l-1}), \quad (1)$$

where $\theta$ is the network parameters to be optimized. Following the approach of LLaVA-Med, we conduct training in two stages. The first stage involves aligning ultrasound images with corresponding medical concepts using simple instructions. In the second stage, we utilize diverse instructions generated by GPT-3.5 [23] to enable our model to address free-form conversations. During training, we freeze the visual encoder and most of the LLM weights and update the parameters of the projection layers, LoRA [10], and our designed modules for enhancement. Through iterative optimization, the model assimilates a substantial volume of new ultrasound visual information and aligns it with medical textual concepts. As a result, it can serve as an ultrasound visual chatbot.

### 3.2 Visual Enhancement

Most existing MLLMs utilize the CLIP series [7, 29, 33] as visual modules, incorporating features from deep layers that represent the global context as inputs to the LLM. However, it may result

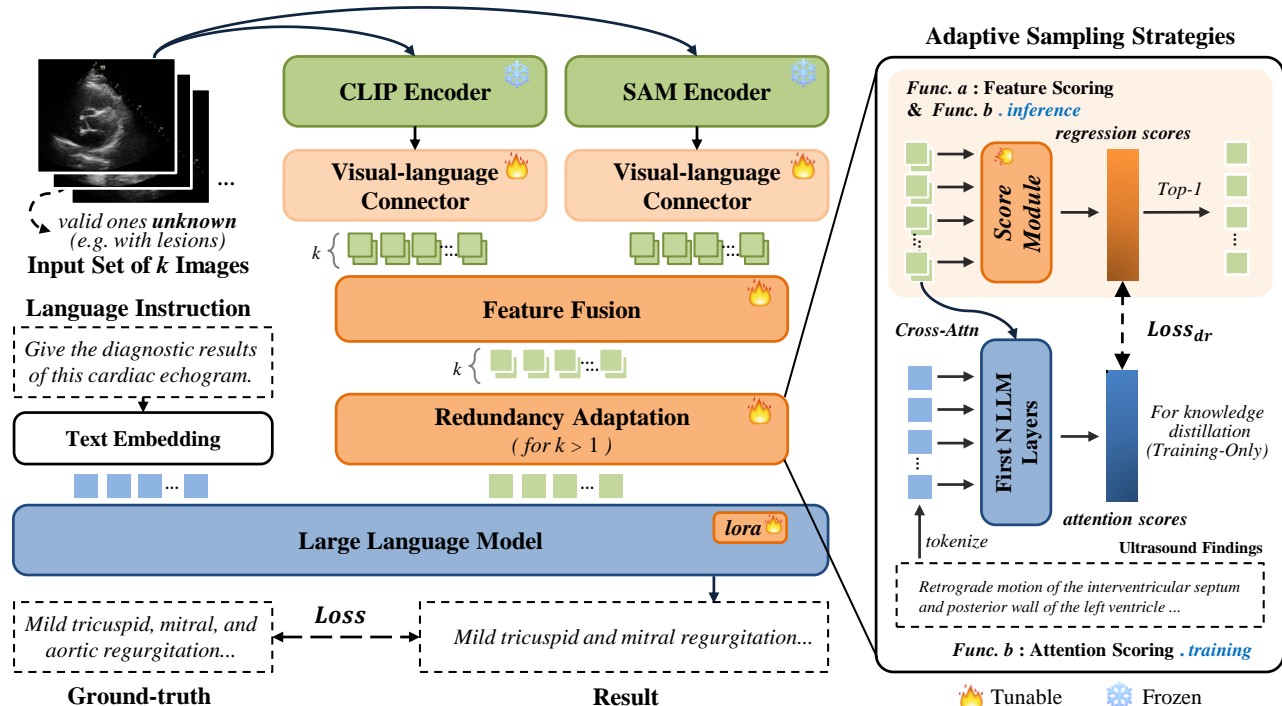

**Figure 2: Overview of our proposed LLaVA-Ultra. Beyond employing the conventional MLLM's architecture, it achieves visual enhancement via a fusion module to incorporate fine-grained SAM features. Additionally, our model can adapt to the data redundancy commonly occurring in medical scenarios by two designed automatic sampling strategies.**

in less fine-grained visual perception in MLLMs. Additionally, the improvement of MLLMs is hindered by the limitations of the visual branch [19, 30, 42], primarily due to the unbalanced scales of visual and language models (e.g. ViT-Large-300M vs. LAMA-7B/13B) [13]. Therefore, there is a need for visual enhancement in MLLMs, especially in the medical domain where image information is subtle. To address this issue, we integrate the Segment Anything Model (SAM) [16], effective in capturing finer-grained features, as an extra visual encoder and further incorporate it through a fusion strategy, as shown in Fig. 2. Specifically, the input images undergo processing by both the CLIP and SAM (ViT-Large-based) encoders $F_1$, $F_2$ to derive visual features and then align them with the linguistic feature space of the LLM through corresponding projection modules $P_{\theta_1}$, $P_{\theta_2}$. We combine the resultant features $H_1$, $H_2$ using a learnable weight parameter $\alpha$ for feature fusion as follows,

$$H_1 = P_{\theta_1}(F_1(X_v)), H_2 = P_{\theta_2}(F_2(X_v)),$$
$$H_v = \alpha \cdot H_1 + (1 - \alpha) \cdot H_2. \tag{2}$$

It enables an appropriate balance between the two typical visual features. The fused feature $H_v$ enriches with detailed local information, such as the texture of lesion areas. It is subsequently concatenated with the instruction tokens to serve as input for the LLM, thereby enhancing the fine-grained visual perception of the MLLM.

**Discussion.** (*i*) *Medical domain adaptation.* This is particularly crucial for medical images, where features like lesion areas are often more subtle compared to natural images. (*ii*) *Parameter-efficient scheme.* It validates the feasibility of visual enhancement

in a parameter-efficient way for multiple frozen visual encoders. (*iii*) *Model extension.* In addition to SAM, it can be generalized to explore the application value of other vision models for MLLMs.

### 3.3 Adaptive Sampling for Data Redundancy

Data redundancy is commonly encountered in clinical scenes, where there is a group of images corresponding to the same text but only some images are valid. To effectively deal with this scenario while balancing computational cost, we devise an adaption module. For such a paired instance, we calculate the weight scores of the grouped $k$ images based on the features obtained from the visual-language projection. Then we sample the image with the highest score as the valid one that best matches the specifics of the text. Specifically, we design two adaptive sampling strategies, as shown in Fig. 2:

(*a*) **Feature scoring strategy.** We use the projected image feature $H_v$ as the weight score, as the optimization of the projection modules is related to the image-text alignment during training. Since $H_v$ contains multiple tokens $h$, each focusing on different content, we avoid treating them equally, such as by simply summing or averaging. Instead, we employ a set of learnable parameters $w$ to calculate a weighted average as a score. And we sample the image with the highest score as the valid one to utilize in this group, *i.e.*,

$$s^i_{fea} = \sum_{j=1}^{d} w_j \cdot h_j,$$
$$s^u_{fea} = Sampling(\{s^i_{fea}\}), i = 1, 2, \ldots, k, \tag{3}$$

where $d$ is the number of channels in feature $H_v$. Thus, these weights can be progressively optimized during training to focus on tokens that are more expressive of the relevance of the image to the text.

(*b*) **Attention scoring strategy.** The strategy above indirectly learns to align during training. However, in our dataset, each text instance contains sufficient information to directly match the text with the most suitable image by comparing the text with each image's features. For instance, if the instruction's question pertains to the diagnosis, we can utilize the descriptive textual ultrasound findings. We leverage the first $N$ LLM layers to perform cross attention in Visual Transformer (ViT) [6, 36] of each image feature in the paired instance and the same text of ultrasound findings $H_t$, *i.e.*,

$$s_{attn}^i = Attn(H_v^i, H_t), i = 1, 2, \ldots, k. \tag{4}$$

The obtained scores directly reflect the relevance of each image to this same text. Considering the absence of additional textual knowledge during inference, we treat these attention scores as pseudo labels and calculate the cross-entropy loss with feature scores $s_{fea}$ above after normalization, *i.e.*,

$$L_{adr} = -\sum_{i=1}^{k} Norm(s_{attn}^i) \log(Norm(s_{fea}^i)). \tag{5}$$

By optimizing the weight parameter $w$ with this knowledge, we can better prioritize tokens and thus screen the image that strongly correlates with the text.

**Discussion.** (*i*) *Fitting medical scenarios.* Our method leverages redundant medical data, which is highly relevant and practical for physicians' diagnostic processes in real-world scenarios. (*ii*) *Computational efficiency.* Instead of captioning all images corresponding to the same text and selecting the most accurate results, we offer a computationally low-cost approach. We directly select feature scores or use a small-scale attention module to screen effective images before feeding them into the LLM. Even with the employment of the redundancy adaptation module and the additional visual encoder mentioned above, it only takes 60 hours on 4 48G A40s for training. (*iii*) *Data utilization.* In the second screening method above, we leverage the textual information of ultrasound observations, a previously untapped resource. It often directly reflects the image's information more intuitively than diagnostic results, thus better identifying valid images. Additionally, the ultrasound observations text and the subsequent diagnostic results input into the LLM are strongly interconnected. The former serves as the surface-level description, while the latter offers a summary and in-depth characterization of the former. This intrinsic relationship facilitates the extraction of more detailed and profound medical semantics during learning. (*iv*) *Domain adaptation.* Although the focus of this paper is on the ultrasound domain, it can be generalized to other medical imaging modalities, such as CT, CXR, and MRI. There are also requirements for fine-grained analysis and cases of data redundancy in these domains. It is possible to utilize features and knowledge of these domains to construct the corresponding assistants similar to our proposed method.

## 4 PROFESSIONAL ULTRASOUND MULTI-MODAL DATA

### 4.1 Ultrasound Multi-modal Data

There is a lack of Chinese medical datasets to perform finetuning for MLLMs. To fill this gap, we present a first attempt to utilize a large-scale Chinese multimodal ultrasound hospital dataset and it has the noteworthy following characteristics: (*i*) *First-hand source and diversity.* Our dataset is directly sourced from the hospital database. It consists of over 20k medical text descriptions paired with 170k ultrasound images with more than 20 examination sites, such as heart, thyroid, breast, uterus, prostate, etc. Such large-scale first-hand Chinese medical data has rarely been achieved in previous works. (*ii*) *Professionality.* It contains comprehensive and detailed clinical text such as examination sites, medical histories, ultrasound observations, diagnosis, etc. Professional doctors offer all the content, ensuring data reliability, which is rarely realized in prior datasets. It makes our work a valuable contribution to applying clinical information to scientific research. (*iii*) *Challenges within ultrasound modality.* Medical imaging modalities used in existing models typically include chest X-ray (CXR), computed tomography (CT), and magnetic resonance imaging (MRI). It's relatively possible for even a layman to identify the body parts in these images, such as the head and chest. However, when confronted with ultrasound, its imaging characteristics make it difficult for non-physicians to achieve this task. This inherent professional barrier poses a greater challenge for MLLMs to learn medical semantics like a layman. (*iv*) *Fine granularity.* Our data contains many samples with high similarity due to the hospital source, a feature frequently absent in existing medical datasets. Thus, it places a higher requirement on fine-grained medical understanding. (*v*) *Respect for medical reality.* Datasets like PMC-15M used by LLaVA-Med typically feature paired instances where a single image corresponds to a text. However, it is often inconsistent with clinical practice where data redundancy exists. Our first-hand dataset addresses this by capturing many instances where one text is paired with multiple images stemming from frames in the same ultrasound video. For example, when the text describes a lesion, only images of frames scanned to the lesion are considered valid for mirroring the text, while those without lesion display are invalid. Yet it lacks intuitive labels for validity determination. Hence, it challenges the model to distinguish valid images for reasoning and holds practical relevance.

### 4.2 Ultrasound Instruction-following Data

Inspired by LLaVA-Med, we generate Chinese ultrasound instruction-following data as Fig. 3. For one caption $X_c$ paired with $k$ images $X_v$, we create an instruction-following example with question $X_q$:

Human : $X_q$ $X_v^1 \ldots X_v^k$<STOP>\n Assistant : $X_c$<STOP>\n

Due to the diverse types of captions and sampled questions, the instructions require answers related to examination sites, ultrasound observations, or medical diagnoses. To validate data rationality, we produce two versions of instruction data: (*i*) Our ultrasound dataset that considers examination sites as cues in questions. (*ii*) A dataset of similar size without sites mentioned in questions, while expecting the model to state it in answers. They are utilized in experiments to evaluate their impact on trained LLaVA-Ultra.

**Ultrasound Multimodal Data Example**

**Examination Site:**

Cardiac color Doppler ultrasound, left heart function measurement, ventricular wall motion analysis, tissue Doppler imaging

**Ultrasound Observation :**

EF: 67.3% AAO: 36.9 mm AO: 33.8 mm LA: 31.1 mm PA: 23.8 mm No significant abnormalities in aortic and pulmonary valve morphology, echogenicity, or activity were noted. The internal diameter of each atrium was normal. The morphology, echo, and activity of the mitral and tricuspid valves were not significantly abnormal. The interventricular septum and the posterior wall of the left ventricle showed retrograde motion. The left ventricular wall activity was not abnormal. There was no continuous interruption of the cardiac structures, and no obvious liquid dark areas were seen in the pericardial cavity.CDFI: A small amount of regurgitant signal was seen in the tricuspid, mitral, and aortic orifices.TDI: Diastolic E/A at the level of the mitral annulus was >1.0.

**Diagnosis :**                                              *Source: hospital*
                                                             *Language: Chinese*
Mild tricuspid, mitral, and aortic regurgitation

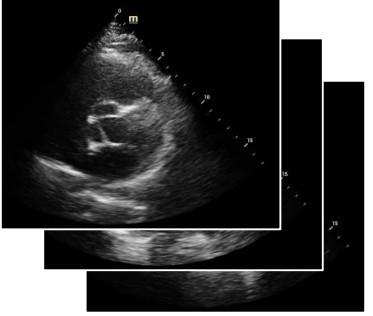

*Data redundancy:*
*a single **text** instance (left) paired*
*with **k** images, only some are valid*
*(e.g. display lesions mentioned in text )*

**Visual Instruction-Following Conversations:**

*User:* What body part is this ultrasound image from and what test is it?

*Assistant:* It is a cardiac color Doppler ultrasound, left heart function measurement, wall motion analysis, and tissue Doppler imaging.

*User:* Describe the condition of the heart in this ultrasound image.

*Assistant:* In this ultrasound image, all structures of the heart are normal, including the aortic valve, pulmonary valve, and mitral and tricuspid valves. The septum and posterior wall of the left ventricle are in retrograde motion, and there is no abnormal left ventricular wall motion. There were no discontinuous structures, and there were no obvious dark areas of fluid in the pericardial cavity.

*User:* What about the morphology and activity of the aortic and pulmonary valves?

*Assistant:* The morphology and activity of the aortic and pulmonary valves are within normal limits.

*User:* Can you describe the regurgitant signal?

*Assistant:* On CDFI, a small amount of regurgitant signal can be observed in the tricuspid, mitral and aortic orifices.

*User:* Please give a diagnosis based on this information.

*Assistant:* Based on the ultrasound findings, the diagnosis is mild tricuspid, mitral and aortic regurgitation.

**Figure 3: An example of our GPT-3.5 [23] generated instruction-following data. Top: A professional multimodal instance from our Chinese ultrasound hospital dataset. There exists data redundancy where a text corresponds to multiple images, but only those mirroring textual descriptions are valid (e.g., display lesions mentioned in the text). Bottom: The instruction-following data generated by GPT-3.5 using the textual descriptions.**

## 5   EXPERIMENTS

### 5.1   Implementation Details

**Datasets.** Besides our large-scale hospital ultrasound dataset, we evaluate models on two open-source medical visual question and answer (Med-VQA) datasets, as illustrated in Tab. 1: (*i*) **SLAKE** [20] is an English-Chinese bilingual dataset and contains 642 images and over 7000 Q&A pairs covering 12 diseases and 39 organs, with CT, MRI, and X-Ray imaging modalities. The Q&As span various topics such as diagnosis, anatomical structure, and lesion location. Presently, SLAKE serves as a crucial benchmark for VQA tasks in the medical domain for its diversity. It has been utilized for evaluation purposes in significant medical multimodal large language model works, such as LLaVA-Med, Med PaLM M, and PMC-VQA. (*ii*) **OpenI** [5, 12] is a chest radiograph dataset from Indiana University Hospital. XrayGLM [40] has preprocessed these unstructured data and translated the English reports into Chinese using ChatGPT. This can provide support in the case of insufficient open-source Chinese multimodal medical data. Our experiments adopt this obtained set comprising 6,436 images and 3,218 reports.

**Evaluation Metrics.** Based on LLaVA-Med's evaluation metrics, we've made further improvements. We tokenize model-generated answers and ground-truth separately, then calculate metrics. For open-set questions, we report the Exact Match (EM) score, F1 score, Precision, Recall and Bilingual Evaluation Understudy (BLEU) score [25]. Notably, BLEU score utilizes precise matching of 4-grams of sequences to evaluate text generation results. We adjust the weights of n-grams and obtain the scores with 1-gram, 2-gram, 3-gram and the uniform set for comprehensive evaluation in accuracy and fluency. In SLAKE, we extra report LLaVA-Ultra's results in CT, MRI, and X-Ray subsets and utilize accuracy scores for the closed-set.

**Experiment Details.** LLaVA-Ultra is initialized with weights from LLaVA-Med in modules included in the latter. It utilizes CLIP-ViT-L/14 and extra SAM-ViT-L as visual encoders and LLaMA-13B as LLM. We use the linear projection for multimodal connection. To ensure a fair comparison, we maintain parameters of compared baseline models in accordance with the respective papers or codes and train until loss functions converge. Data preprocessing ways are also the same. Experiments employ 4 48GB NVIDIA A40s with PyTorch. Models are optimized by Adam [15] with learning rate $1e^{-3}$ and batch size 16.

**Table 1: Datasets for evaluation. For the bilingual dataset, the content of both language versions is essentially identical. Therefore, we present only the data attributes of the Chinese version here.**

| | US-Hospital | | | SLAKE | | | OpenI | | |
|---|---|---|---|---|---|---|---|---|---|
| Image Modality | Ultrasound | | | CT, MRI, X-Ray | | | Chest X-Ray | | |
| Subset | Train | Val | Test | Train | Val | Test | Train | Val | Test |
| Q&A Pairs | 353690 | 11516 | 11346 | 5967 | 607 | 491 | 5836 | 200 | 400 |
| Images | 1604673 | 49994 | 49996 | 542 | 50 | 50 | 5836 | 200 | 400 |
| Text | 176845 | 5758 | 5673 | 5967 | 607 | 491 | 2918 | 100 | 200 |
| Average Word Length | 133 | | | 3 | | | 90 | | |
| Open / Closed | ✓ / ✗ | | | ✓ / ✓ | | | ✓ / ✗ | | |
| CHN / EN | ✓ / ✗ | | | ✓ / ✓ | | | ✓ / ✓ | | |

**Table 2: Quantitative comparisons with prior SOTA methods fine-tuned on relevant datasets. Our model performs best or equal in the Chinese cases, demonstrating its effectiveness and robustness. Its results are acceptable in the English cases that are not included in our Chinese-only pre-training. It indicates the knowledge of our LLM at initialization is not significantly corrupted.**

| Method | EM | F1 | Precision | Recall | BLEU | BLEU-1 | BLEU- 2 | BLEU-3 | Closed-Set Accuracy |
|---|---|---|---|---|---|---|---|---|---|
| *on US-Hospital Chinese dataset* | | | | | | | | | |
| LLaVA | 59.50 | 66.97 | 76.65 | 64.78 | 39.50 | 48.29 | 42.30 | 38.83 | - |
| LLaVA-Med | 50.55 | 70.77 | 73.91 | **73.42** | 40.42 | 50.32 | 43.23 | 39.36 | - |
| LLaVA-Ultra | **62.00** | **72.17** | **78.84** | 72.67 | **48.62** | **57.71** | **51.51** | **47.95** | - |
| *on SLAKE-zh dataset* | | | | | | | | | |
| LLaVA | 72.56 | 75.36 | 76.76 | 76.00 | 44.85 | 70.43 | 62.43 | 53.28 | 65.29 |
| LLaVA-Med | 75.02 | **77.08** | 78.98 | **77.07** | 49.31 | 72.61 | 65.09 | 56.96 | 73.58 |
| LLaVA-Ultra (CT) | 83.34 | 68.96 | 77.13 | 65.39 | 40.91 | 68.49 | 61.58 | 50.54 | 74.50 |
| LLaVA-Ultra (MRI) | 93.39 | 81.18 | 84.64 | 77.36 | 55.07 | 86.99 | 82.09 | 72.23 | 75.16 |
| LLaVA-Ultra (X-Ray) | 90.53 | 80.54 | 83.97 | 76.83 | 65.19 | 86.97 | 79.82 | 72.09 | 83.84 |
| LLaVA-Ultra (All modalities) | **88.99** | 76.85 | **81.88** | 73.15 | **53.95** | **80.76** | **74.39** | **64.90** | **76.77** |
| *on SLAKE-en dataset* | | | | | | | | | |
| LLaVA | 76.75 | 75.83 | 77.20 | 75.99 | 15.47 | 72.95 | 61.18 | 37.54 | 65.50 |
| LLaVA-Med | 78.22 | **77.55** | **78.61** | **78.01** | **18.12** | **74.77** | **62.65** | **39.67** | 65.50 |
| LLaVA-Ultra (CT) | 77.57 | 69.09 | 71.87 | 68.61 | 19.26 | 71.21 | 53.07 | 30.02 | 51.43 |
| LLaVA-Ultra (MRI) | 73.60 | 60.80 | 67.44 | 58.15 | 8.18 | 58.93 | 37.35 | 15.93 | 64.34 |
| LLaVA-Ultra (X-Ray) | 94.67 | 80.07 | 87.21 | 77.10 | 19.66 | 80.57 | 70.06 | 43.99 | 86.27 |
| LLaVA-Ultra (All modalities) | **81.33** | 68.69 | 74.82 | 66.31 | 13.95 | 68.31 | 51.02 | 27.77 | **67.25** |
| *on OpenI-zh dataset* | | | | | | | | | |
| LLaVA | 37.36 | 72.90 | 76.78 | 71.06 | 25.36 | 48.29 | 28.42 | 20.62 | - |
| LLaVA-Med | 35.13 | **74.73** | 79.26 | **71.96** | 27.10 | 50.60 | 30.44 | 22.10 | - |
| LLaVA-Ultra | **49.32** | 72.71 | **82.83** | 70.43 | **29.37** | **51.81** | **32.61** | **24.40** | - |
| *on OpenI-en dataset* | | | | | | | | | |
| LLaVA | 41.63 | 56.36 | 61.58 | 54.24 | 17.50 | 39.45 | 21.77 | 13.67 | - |
| LLaVA-Med | **46.78** | **57.23** | **63.44** | 54.10 | **18.50** | 39.85 | 22.54 | **14.65** | - |
| LLaVA-Ultra | 41.90 | 56.75 | 61.10 | **55.16** | 18.49 | **40.31** | **22.67** | 14.56 | - |

## 5.2 Performance and Comparisons

We provide the comparisons between LLaVA, LLaVA-Med, and our LLaVA-Ultra when employed as a medical visual chatbot.

For qualitative comparisons shown in Fig. 4, LLaVA model for the general domain struggles with medical tasks, highlighting the gap between domains. While tailored for the medical domain, LLaVA-Med still inadequately addresses Chinese ultrasound scenes. It often focuses solely on textual medical concepts in the questions and offers vague and invalid answers that do not meet the questioner's needs. They tend to rely more on medical knowledge learned early from the LLMs rather than effectively incorporating medical visual features. This results in responses that display weak correlations with the input images and even exhibit inaccuracy. It implies the flaws in their model structures. In contrast, our LLaVA-Ultra model

**Examination Site:**
Cardiac color Doppler ultrasound, left heart function measurement, ventricular wall motion analysis, tissue Doppler imaging

**Diagnosis:**
Mild tricuspid, mitral, and aortic regurgitation

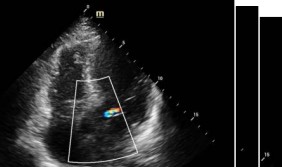

**Observation:**
EF: 67.3% AAO: 36.9 mm AO: 33.8 mm LA: 31.1 mm PA: 23.8 mm No significant abnormalities in aortic and pulmonary valve morphology, echogenicity, or activity were noted. The internal diameter of each atrium was normal. The morphology, echo, and activity of the mitral and tricuspid valves were not significantly abnormal. The interventricular septum and the posterior wall of the left ventricle showed retrograde motion. The left ventricular wall activity was not abnormal. There was no continuous interruption of the cardiac structures, and no obvious liquid dark areas were seen in the pericardial cavity. CDFI: A small amount of regurgitant signal was seen in the tricuspid valve orifice, mitral valve orifice, and aortic valve orifice.TDI: Diastolic E/A >1.0 at the level of the mitral annulus.

**Examination Site:**
Ultrasound of Liver, Gallbladder, Pancreas and Spleen

**Diagnosis:**
Fatty liver; liver cysts

**Observation:**
The liver was full in shape, with a smooth envelope, and a cystic dark area was seen in the liver, about 0.70 cm in size, with a thin and smooth wall, good internal translucency, and enhanced posterior echogenicity; the rest of the parenchymal echogenicity was enhanced, dense, and poorly distributed, and the vascular network was poorly defined. The left and right intrahepatic bile ducts were not significantly dilated. The shape and size of the gallbladder were normal, the wall of the gallbladder was smooth without thickening, the bile in the lumen was well transmissible, No abnormal echoes were seen inside the gallbladder. The internal diameter of the common bile duct was not significantly widened, and there was no obvious abnormality in the lumen. The spleen was normal in shape and size, and the echoes in the spleen were uniform and small. The pancreas was normal in shape and size, with homogeneous internal echogenicity, no obvious dilatation of the main pancreatic duct, and no obvious foci of occupancy.

---

**User:** *Where in the body does this ultrasound image come from?*

**LLaVA-Ultra:** This ultrasound is from color Doppler ultrasound, left heart function measurement, ventricular wall motion analysis, tissue Doppler imaging

**LLaVA-Med:** This ultrasound is from an ultrasound scan of the heart.

**LLaVA:** This ultrasound image is from a heart exam.

**User:** *Where in the body does this ultrasound image come from?*

**LLaVA-Ultra:** It is from ultrasound of liver, gallbladder, pancreas and spleen

**LLaVA-Med:** It is from a pregnancy test, specifically a fetal ultrasound.

**LLaVA:** This ultrasound is from the stomach.

---

**User:** *What information does this ultrasound of cardiac color Doppler ultrasound, left heart function measurements, ventricular wall motion analysis, and tissue Doppler imaging show?*

**LLaVA-Ultra:** The heart had normal internal diameters in all atria, normal internal diameter of the left atrium, and no significant abnormalities in the morphology, echogenicity, or activity of the aortic and pulmonary valves. The internal diameter of each atrium was normal. The morphology, echo, and activity of the mitral and tricuspid valves were not significantly abnormal. The thickness of the interventricular septum and the posterior wall of the left ventricle was normal with retrograde motion. No abnormal left ventricular wall activity was seen, EF: 65%. There was no continuous interruption of the cardiac structures, and no obvious liquid dark areas were seen in the pericardial cavity. CDFI:A small amount of regurgitant signal was seen in the tricuspid orifice and mitral orifice. TDI:Diastolic E/A at the level of the mitral annulus was <1.0.

**LLaVA-Med:** The image display shows an ultrasound of the heart showing the left ventricle and aorta and is color-coded to indicate the electrical activity of the heart. This visualization provides valuable information about heart function and helps physicians analyze and diagnose any potential problems or abnormalities. The image also includes graphs showing heart rate variability, which can be an important indicator of overall cardiovascular health. If a computer monitor screen is present in the scene, it indicates that this information is being displayed and analyzed by a medical professional or researcher.

**LLaVA:** This diagram of cardiac color Doppler ultrasound, left heart function measurement, ventricular wall motion analysis, shows the activity and function of the heart. It can help your doctor understand the condition and function of the heart, as well as assess the health of the heart.

**User:** *What does this ultrasound of a liver, gallbladder, pancreas, and spleen ultrasound show?*

**LLaVA-Ultra:** The shape and size of the liver were within normal range, the peritoneum was smooth. A cystic dark area was seen in the liver, the vascular network was not clear, and the left and right intrahepatic bile ducts were not obviously dilated. The shape and size of the gallbladder were normal, the wall of the gallbladder was smooth without thickening, the bile in the lumen was well transmissible, and no obvious abnormal echoes were seen in the gallbladder. The internal diameter of the common bile duct was not significantly widened, and there was no obvious abnormality in the lumen. The spleen was normal in shape and size, and the echoes in the spleen were uniform and small. The pancreas was normal in shape and size, with homogeneous internal echoes, no obvious dilatation of the main pancreatic duct.

**LLaVA-Med:** The image is an ultrasound image of the liver showing its different lobes in what appears to be a three-dimensional view. This type of ultrasound scan provides detailed information about the structure and function of the liver and can be used for diagnostic purposes. The image does not show any abnormalities in the liver or its blood vessels, which indicates that the liver appears to be healthy. The presence of the gallbladder suggests that the bile ducts are functioning properly and the overall image provides a clear view of the normal anatomy and physiology of the liver.

**LLaVA:** This diagram of a liver, gallbladder, pancreas, and spleen ultrasound shows some medical information, including the shape and location of the liver, gallbladder, pancreas, and spleen. This information can help doctors diagnose and treat a patient's disease, as well as assess a patient's health.

---

**User:** *Give the medical diagnosis of this ultrasound image.*

**LLaVA-Ultra:** Mild tricuspid, mitral, and aortic regurgitation

**LLaVA-Med:** A color image showing the left ventricle of the heart, part of the electrocardiogram, along with a fetal heartbeat monitor. This indicates that the image is relevant for medical diagnosis and heart health monitoring. The electrical activity of the heart is displayed in a polar plot.

**LLaVA:** The medical diagnosis of this diagram of cardiac color Doppler ultrasound, left heart function measurements, ventricular wall motion analysis, and tissue Doppler imaging may be to assess the function and health of the heart. Doppler ultrasound is a noninvasive cardiac test that provides information about the structure of the heart. Left heart function measurement measures function of heart's circulation to assess normalcy and function of the heart.

**User:** *Give the medical diagnosis of this ultrasound image.*

**LLaVA-Ultra:** Fatty liver

**LLaVA-Med:** The image is an ultrasound scan of a fetal examination, showing the fetal pole, placenta, and umbilical cord.

**LLaVA:** Based on this picture of liver, gallbladder, pancreas, and spleen ultrasound, the medical diagnosis may include the following: 1. Liver: possible hepatitis, cirrhosis, and hepatic fibrosis. 2. gallbladder: possible cholecystitis, gallbladder stones, and other diseases. 3. pancreas: possible pancreatitis, pancreatic cancer, and other diseases. 4. spleen: possible splenitis, splenic carcinoma, and other diseases. It should be noted that this is only a preliminary medical diagnosis.

**Figure 4: Comparisons in medical visual conversations. LLaVA and LLaVA-Med tend to give** **vague** **answers irrelevant to images and** **wrong** **results. In contrast, LLaVA-Ultra offers more** **correct** **and specific responses associated with visual contents.**

delivers more correct and specific answers closely aligned with the visual content of the input medical image. This notable performance demonstrates the capability of LLaVA-Ultra.

Table. 2 further presents the quantitative comparison results. We finetune LLaVA and LLaVA-Med on the training set of our Chinese ultrasound hospital dataset and report their metrics on the test set. For SLAKE and OpenI, we finetune and evaluate all three

**Table 3: Ablation study: several variants of our model structure on US-Hospital Chinese dataset.**

| Variant | EM | F1 | Precision | Recall | BLEU | BLEU-1 | BLEU-2 | BLEU-3 |
|---|---|---|---|---|---|---|---|---|
| w/o. Visual enhancement | 57.01 | 71.37 | 75.23 | 72.11 | 45.81 | 55.96 | 49.15 | 45.12 |
| w/o. Redundancy adaption | 50.13 | 70.57 | 72.37 | **73.91** | 41.03 | 51.57 | 44.24 | 40.08 |
| w/o. Func.b redundancy adaption | 51.86 | 70.95 | 73.75 | 73.84 | 41.52 | 51.09 | 44.21 | 40.39 |
| w/o. Stage 2 tuning | 54.52 | 71.36 | 75.83 | 72.45 | 43.54 | 53.19 | 46.36 | 42.59 |
| w/o. Site prompt in instruction | 57.48 | 71.57 | 76.84 | 71.63 | 45.13 | 54.61 | 48.03 | 44.31 |
| LLaVA-Ultra full model | **62.00** | **72.17** | **78.84** | 72.67 | **48.62** | **57.71** | **51.51** | **47.95** |

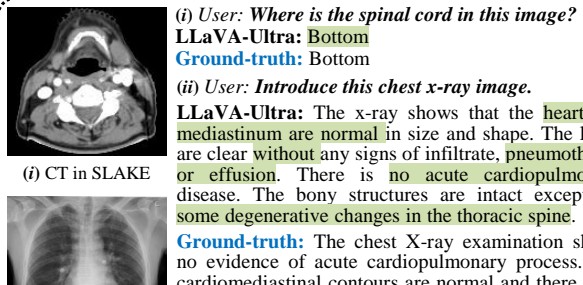

*(i) User: **Where is the spinal cord in this image?***
**LLaVA-Ultra:** Bottom
**Ground-truth:** Bottom

*(ii) User: **Introduce this chest x-ray image.***
**LLaVA-Ultra:** The x-ray shows that the heart and mediastinum are normal in size and shape. The lungs are clear without any signs of infiltrate, pneumothorax, or effusion. There is no acute cardiopulmonary disease. The bony structures are intact except for some degenerative changes in the thoracic spine.
**Ground-truth:** The chest X-ray examination shows no evidence of acute cardiopulmonary process. The cardiomediastinal contours are normal and there is no focal consolidation, pleural effusion or pneumothorax identified. However, degenerative changes of the thoracic spine are observed. The patient also has right greater than left biapical bullous emphysema.

*(i)* CT in SLAKE

*(ii)* X-Ray in OpenI

**Figure 5: Case study: downstream English tasks reveals Chinese pretrain does not visibly damage early LLM knowledge.**

models. The results clearly show that LLaVA-Ultra outperforms both LLaVA and LLaVA-Med on our Chinese ultrasound dataset. It proves the superiority of our model architecture, particularly the visual enhancement and redundancy adaptation modules. When evaluated on the downstream tasks of Chinese language in SLAKE and OpenI, LLaVA-Ultra consistently delivers the best performance across open-set and close-set questions, which demonstrates its robustness. This suggests that our model effectively learns to map medical semantics to correct features during training. Even on English data subsets, LLaVA-Ultra offers acceptable results shown in Table. 2 and Fig. 5, which may benefit from its textual knowledge acquired during LLM initialization. This indicates that its training on the Chinese dataset does not significantly compromise this part of English knowledge [46]. We also report results in CT, MRI, and X-Ray subsets on SLAKE besides the overall average values. Our model pretrained on ultrasound modality shows effective adaptability across diverse imaging modalities. It highlights LLaVA-Ultra's robustness and versatility in accomplishing downstream multimodal tasks after simple finetuning.

### 5.3 Ablation Study

To assess the validity of the model's components, we conducted comparative experiments on several aspects shown in Tab. 3.

**Visual enhancement.** To prove the necessity of visual enhancement, we replace our dual visual encoders and their fusion module with the original single CLIP encoder and observe the decrease of all metrics in Tab. 3. This verifies the necessity of strengthening the visual branch of MLLMs and underlines the effectiveness of our

feature fusion strategy. The integration of SAM features enables LLaVA-Ultra to extract finer-grained visual semantics, which is a crucial aspect of handling subtle information in medical scenarios.

**Data redundancy adaptation.** We remove our data redundancy adaptation module, the metrics in Table. 3 show a significant reduction. This highlights the importance of addressing data redundancy, which is prevalent in real medical scenarios but less noticed. In previous works, when multiple images correspond to the same text, such text is often assigned to each image. It results in mapping the images that don't mirror the specific text, i.e. redundant data, and the valid images to a similar feature representation. This hinders the model from learning accurate medical semantics and cross-modal alignment. As shown in the scores, LLaVA-Ultra full model can address this issue effectively with our adaption strategy. Specifically, our attention scoring strategy (Func.b) leverages rich textual data for feature alignment and thus achieves better scores compared to the simpler feature scoring (Func.a).

**Data construction.** We modified the instruction data by removing the cues of the examination site from the question. Instead, we ask a generalized question like *"give the diagnosis of this ultrasound image"*. Results in Tab. 3 indicate a slight decrease in evaluation metrics within acceptable limits. This demonstrates the effectiveness of limited cues for the model to learn specific medical knowledge, as well as the robustness of our model structure.

### 5.4 Limitation

Although LLaVA-Ultra shows impressive capabilities in Chinese medical multimodal understanding, it still has some limitations, including: 1) Its performance is hindered by the scale of the pre-trained vision models. 2) Our large-scale medical dataset has not yet included more comprehensive labels e.g., segmentation to allow our model to engage further enhancement in visual perception.

### 6 CONCLUSION

We propose LLaVA-Ultra, a large Chinese language and vision assistant for the ultrasound domain. To achieve this, we create a high-quality visual-language instruction-following dataset from the large-scale professional ultrasound database of hospitals. More importantly, we improve the conventional visual language model structure by performing visual enhancement and data redundancy adaptation. It enables LLaVA-Ultra to fit the needs of fine-grained medical information and practical clinical scenarios, thus producing high-quality responses in medical visual conversations. On three medical VQA datasets, LLaVA-Ultra outperforms previous SOTA in various metrics, demonstrating its effectiveness and robustness.

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
