# OpenReview forum: "LLaVA-Ultra: Large Chinese Language and Vision Assistant for Ultrasound"
_acmmm.org/ACMMM/2024/Conference — MM2024 Poster_

### Official Review · Reviewer_51Hx · 2024-05-08

**Rating:** 5
**Confidence:** 3

**Summary:**

This paper present LLaVA-Ultra, a LVM assistant for ultrasound understanding. LLaVA-Ultra adopts similar methods to LLaVA to bring the ultrasound images into training but also introduces new domain-specific modules such as the redundancy adaptation. The model was trained by 170k ultrasound images paired with 20k clinical texts from an in-house dataset, which is further processed by GPT-3.5 to produce instruction-following conversations like LLaVA-Med. The proposed model showed improved performance on several benchmarks and the authors also presented interesting analyses comparing different models.

**Strengths:**

1. Overall, multi-modal learning for medical images is an interesting and important task.
2. A new model, LLaVA-Ultra, is proposed with some novel domain-specific components in addition to the vanilla LLaVA.
3. A potentially useful dataset is also contributed.
4. Writing is mostly clear.

**Limitations:**

1. The evaluations only use automatic metrics. However, they might not correlate well with expert evaluations. As such, the authors should conduct larger-scale human evaluation on the new model.
2. It is not clear whether the dataset will be publicly available.
3. The compared baselines are pretty basic without proper referring to other studies using these datasets. For example, https://dl.acm.org/doi/abs/10.1145/3581783.3611830 should also be mentioned or compared.
4. The authors should discuss more on how a model trained by ultrasound images is supposed to perform well on medical VQA datasets involving other modalities (e.g., X-ray).
5. It seems suboptimal to use a fixed CLIP/SAM encoder. Have you tried continuing pre-training one encoder using your paired data with contrastive learning?

**Suitability:**

2

---

### Official Review · Reviewer_188y · 2024-05-23

**Rating:** 3
**Confidence:** 3

**Summary:**

This article presents a Large Chinese Language and Vision Assistant for question answering on a Chinese ultrasound medical dataset. The innovation lies in integrating a Segment Anything Model into the visual encoder. In cases where a single text is paired with multiple figures, they employ weighted scoring with knowledge distillation to adaptively screen valid images reflecting text descriptions.

**Strengths:**

- The article addresses a medical visual question-answering task on an untapped Chinese dataset.
- The model shows partial superiority in Chinese performance compared to SOTA.

**Limitations:**

- The article only shows partial superiority in Chinese performance compared to SOTA; in Table 2, it does not surpass SOTA on English datasets.
- The article does not utilize semantics-based evaluation metrics.
- As the instructions are generated by GPT, I wonder about the difficulty of converting the whole dataset into English.
- Which module of your approach is adapted to Chinese?

**Suitability:**

3

---

### Official Review · Reviewer_3Z8y · 2024-05-29

**Rating:** 4
**Confidence:** 4

**Summary:**

The paper presents LLaVA-Ultra, a multimodal large language model specifically designed for Chinese medical visual conversations, particularly focusing on ultrasound images. This model aims to address the limitations of current visual language models in the medical domain by providing more precise and relevant responses to medical visual queries. The model utilizes a fusion module with fine-grained vision encoders and adapts to data redundancy in medical scenarios through weighted scoring and knowledge distillation. The authors demonstrate the effectiveness of LLaVA-Ultra on three medical visual question-answering (VQA) datasets, demonstrating comparable or superior performance to existing models.

**Strengths:**

1. The paper is written clearly and is easy to understand.
2. The research addresses a crucial gap in the field of medical visual language models, specifically in the Chinese medical domain. By developing a model that can accurately interpret and respond to visual medical data, this work has the potential to significantly impact medical diagnostics and education.
3. The authors conducted experiments on three medical visual question-answering datasets, including a large-scale Chinese ultrasound dataset sourced directly from hospitals. The results consistently demonstrate the model’s superior performance compared to previous state-of-the-art models, validating the effectiveness of their approach.

**Limitations:**

1. During model training, both CLIP encoder and SAM encoder are frozen. Since both models are trained on natural images rather than medical images, will this affect model performance?

2. There are too few comparison methods in the experiment. It is recommended to add the comparison with other medical MLLMs, such as Med-PaLM

3. The results in bold in Table.2 are a bit confusing. It would be helpful to clarify the criteria for highlighting these results.

4. The BLEU indicator is a relatively mechanical comparison of the accuracy of n-grams, which is not suitable for evaluating the generation results of LLM. It is recommended to add the evaluation results of metrics such as Meteor and CIDEr.

5. Has LLaVA-Med been fine-tuned on the US-Hospital Chinese dataset? If it has not been fine-tuned, the comparison seems a bit unfair. In other words, how to prove that the model structure of LLAVA-Ultra is better than LLaVA-Med?

**Suitability:**

3

---

### Meta-Review · Program_Chairs · 2024-07-12

**Recommendation:** Accept (Poster)
**Confidence:** 4

**Metareview:**

Thanks for submitting the paper to ACM Multimedia. The reviewers felt that the application area was very interesting and the new approach and the considered dataset can spur future growth in this research space. The reviewers also had questions about the validation approach. Many of the reviewer concerns were allayed with the rebuttal. The paper is inherently multimodal and will generate interest at the conference. An acceptance is recommended.